# Localisation of Intracellular Signals and Responses during Phagocytosis

**DOI:** 10.3390/ijms24032825

**Published:** 2023-02-01

**Authors:** Maurice B. Hallett

**Affiliations:** School of Medicine, Cardiff University, Cardiff CF14 4XN, UK; hallettmb@cf.ac.uk

**Keywords:** phagocytosis, Ca^2+^, phospholipids, cell signalling, neutrophils, cytoskeleton

## Abstract

Phagocytosis is one of the most polarised of all cellular activities. Both the stimulus (the target for phagocytosis) and the response (its internalisation) are focussed at just one part of the cell. At the locus, and this locus alone, pseudopodia form a phagocytic cup around the particle, the cytoskeleton is rearranged, the plasma membrane is reorganised, and a new internal organelle, the phagosome, is formed. The effect of signals from the stimulus must, thus, both be complex and yet be restricted in space and time to enable an effective focussed response. While many aspects of phagocytosis are being uncovered, the mechanism for the restriction of signalling or the effects of signalling remains obscure. In this review, the details of the problem of restricting chemical intracellular signalling are presented, with a focus on diffusion into the cytosol and of signalling lipids along the plasma membrane. The possible ways in which simple diffusion is overcome so that the restriction of signalling and effective phagocytosis can be achieved are discussed in the light of recent advances in imaging, biophysics, and cell biochemistry which together are providing new insights into this area.

## 1. Introduction

Phagocytosis is the process by which a cell internalises a particle, such as a bacterium or fungal cell (usually 0.5–5 μm), by extending pseudopodia which surround the particle and form an intracellular compartment, the phagosome [1,2,3]. This simple description hides the cellular and molecular complexity of the mechanisms of the process. The target “particle” can be a number of natural and unnatural shapes, including spheres, rods, oblate ellipsoids, prolate ellipsoids, elliptical discs, rectangular discs, and even miniature flying saucers (UFOs) [4,5,6], composed of nonbiological material, e.g., coal particles, polystyrene beads, etc. Usually, the particle is opsonised with either antibody molecules which engage Fc receptors or complement molecules (i3Cb) which engage iCb receptors or β2 integrin on the phagocyte surface and generate intracellular signals which trigger the phagocytic process [1,2,3]. These intracellular signals must not simply switch on the process, but must also inform the cell about the cellular location that the process must be restricted to. It would be no good if pseudopodia were formed all around the periphery of the cell, as the limitation of cell surface area would prevent effective capture and the target would escape. Instead, the phagocytic pseudopodia form only at the particle contact site (and nowhere else on the cell surface) so that the phagocytic cup forms effectively such that the pseudopodia can extend around the particle. It may seem obvious that the local response is triggered by the local engagement of receptors which generate localised intracellular signals, which in turn generate a localised cellular response. However, as “intracellular signals” are simply ions or molecules in the cytosol or molecules in the cell membrane, the question reduces to: how can such signals be localised within the cell? Can the diffusion of such “pre-phagocytic signals” be prevented? The evidence for possible answers to these questions is given and discussed in the first part of the review. 

The other important question about phagocytosis relates to the mechanisms of fusion of enzyme-containing granules with the phagosomal membrane but not the plasma membrane. As the phagosomal membrane is part of the plasma membrane until the phagosome is complete, the question arises as to how “fusion” signals are localised to the phagosomal membrane. Similarly, after completion of the phagosome, the nonmitochondrial NADPH is assembled and activated only on the phagosomal membrane and not on the plasma membrane. Again, the question arises as to what these signals are and how these signals are localised to the phagosomal membrane. Although not the main focus of this review, the evidence for possible answers to these questions about the localisation of “post-phagocytic signals” is given and briefly discussed in the second part of the review.

## 2. Diffusion of Chemical Signals

An important concept in understanding the mechanism of localised intracellular signalling is ***diffusion.*** Diffusion is, of course, the net movement of ions or molecules from a zone of higher concentration to one of lower concentration. Thus, if an intracellular signal is generated within the cell at the contact point between the phagocytic target and the plasma membrane of the phagocyte, its intracellular concentration will be higher at that point than elsewhere in the cell and diffusion of the signal molecules away from the point of contact will occur. Diffusion is, of course, the result of the random Brownian motion of individual ions or molecules, causing the initial concentration of signals to be “diluted out” as molecules move away from a source. Although random at an individual ion/molecule level, the net effect occurs in a predictable way (Figure 1a) and the concentration can be calculated at any point distant from the source as it declines with passing time [7]. The decrease in concentration of a molecule depends on its distance from the source of the signal (the contact between the target and the cell), the time elapsed (t), and its diffusion constant (D, μm^2^/sec).

The equations which govern these relationships, especially in two and three dimensions, can look daunting [7], but the main parameter which governs the process is the diffusion constant, D, a measure of “diffusivity” of the molecule or ion in a given environment. A simple, but useful, approach for estimating how a localised focal signal may be restricted is to calculate its “diffusion length” given by:2 √(Dt),
where D is the diffusion constant (μm^2^/s) and t is the time (s) taken (Figure 1b). The concept of diffusion length applies to both ions in solution and to lipids in a membrane.

## 3. Theoretical Basis of Localisation of Ca^2+^ Signalling 

Phagocytosis by opsonised particles is accompanied by Ca^2+^ influx from the extracellular environment generating intracellular Ca^2+^ signalling [8,9]. The earliest report of phagocytosis by neutrophils showed that Ca^2+^ in the medium was an absolute requirement for phagocytosis [10]. However, an intracellular Ca^2+^ signal may not always be obligatory, and there are early reports using nonopsonised particles in which no Ca^2+^ signals were found [11,12]. Indeed, the phagocytic cup can partially form without any Ca^2+^ signal [13,14]. However, when a critical number of opsonin receptors have been engaged (often about 50% of the cup), Ca^2+^ influx is triggered, causing a large cytosolic Ca^2+^ signal [13]. This Ca^2+^ signal accelerates the rate of completion of phagocytosis and probably accounts for the increased phagocytic rate caused by opsonisation observed in cell populations. The flattening and spreading of neutrophils onto surfaces, which may be considered as “frustrate phagocytosis”, is also accompanied by obligatory large Ca^2+^ signals [15,16]. The Ca^2+^ signal can be thought of as being the permissive signal for efficient phagocytosis.

As suggested earlier, in considering the question of whether this Ca^2+^ signal could be localised to the site of phagocytosis, a useful parameter is the “diffusion length” which indicates how far from the source of the signal Ca^2+^ ions would travel by a defined time. The diffusion length can be calculated from knowledge of the diffusion constant, D, as 2√(Dt. The diffusion constant for Ca^2+^ in water, D_Ca_, has been measured as 7.7 × 10^−6^ cm^2^/s [17]. i.e., approximately 800 μm^2^/s. The diffusion length of Ca^2+^ ions after 100 ms (0.1 s) would thus be 2√(800 × 0.1) = 18 μm. As this is larger than the diameter of many phagocytes (e.g., neutrophils are about 10 μm), it suggests that Ca^2+^ ions are unlikely to act as a localised phagocytic signal. However, the diffusion constant for Ca^2+^ (D_Ca_) in extracted cytosol is lower and has been measured, using Ca^45^ at 530 μm^2^/s [18], 223 μm^2^/s in cytosolic extract from Xenopus oocytes [19], or in living cells using fluorescence correlation spectroscopy at 650 μm^2^/s [20]. Thus, in the cytosol, the diffusion length of free Ca^2+^ is about 9.2–16 μm, which is still too large to generate localised Ca^2+^ signals. However, in the cytosol, Ca^2+^ ions bind reversibly to Ca^2+^ buffering agents, such as proteins and other molecules. It is estimated from the association rate (“on rate” or K_on_) and buffer concentration that the lifetime of a free Ca^2+^ ion would be in the order of 10^−5^s [19], meaning that the free unbuffered Ca^2+^ signal would be extremely localised. However, the buffer molecule is also mobile and will diffuse, carrying Ca^2+^ with it until the ion is released (Figure 2). It is, thus, more appropriate to consider the diffusion of buffered Ca^2+^ ions. The measured diffusion constant for buffered Ca^2+^ is higher than for free cytosolic Ca^2+^ and the diffusion length of the buffered Ca^2+^ is extended from d1 to (d1 + d2). For Ca^2+ -^binding proteins with the EF hand-motif (e.g., calmodulin, calbindin, etc.), the dissociation rate (k_off_) is about 10 s^−1^, giving a “lifetime” of the Ca^2+^–buffer complex of 0.1 s (100 ms). The diffusion constant of the buffered Ca^2+^ has been measured in cytosol extract to be 13 μm^2^/s. After 100 ms (when the Ca^2+^ ion is released), the diffusion length is only about 2.3 μm [19]. Assuming that the released Ca^2+^ ion is free for about 1 s before being taken up into immobile Ca^2+^ stores, the “effective range” of buffered Ca^2+^ is about 5 μm [20]. Although these measurements were made using cytosol extract with a Ca^2+^-buffering content of about 300 μM [19], the Ca^2+^-buffering ratio of neutrophil cytosol is similarly high, about 1000-3000:1 [21,22], meaning that for every free Ca^2+^ ion in the cytosol, there are 1000-3000 bound Ca^2+^ ions, so the conclusion can be extrapolated to neutrophil cytoplasm. The diffusion length predicted can also be experimentally confirmed in neutrophils by osmotically lysing Ca^2+^-containing lysosomes to release a small localised Ca^2+^ signal in the cytosol. This is achieved by the use of a dipeptide glycyl-l-phenylalanine 2-naphthylamide (GPN), which is cleaved within cathepsin C-containing lysosomes and is widely used to osmotically rupture the lysosomal membrane and release its Ca^2+^ content into the cytosol. In neutrophils, this procedure generated “Ca^2+^ puffs” with diameters of about 3 μm [23], similar to those expected from the theoretical diffusion length calculation for buffered Ca^2+^. This suggests that the localisation of Ca^2+^ signals during phagocytosis is at least theoretically possible in neutrophils and probably other phagocytes. However, as the influx of Ca^2+^ from the extracellular environment is the important Ca^2+^ source for phagocytosis, understanding the mechanism for generating the Ca^2+^ signal is crucial.

The diffusion length of free Ca^2+^ is very small (d1), as Ca^2+^ ions are rapidly bound to proteins (buffers). However, the buffer also diffuses (d2), carrying the Ca^2+^ ion to be released at a different location. The time between the binding and release of the Ca^2+^ ion depends on the dissociation rate (k_off_ or off rate), during which time the buffered Ca^2+^ travels the distance d2. Ca^2+^-binding proteins have k_off_ rates of about 10 s^−1^, giving a “lifetime” of 0.1 s. Hence, the diffusion length of the buffered Ca^2+^ is extended from d1 to (d1 + d2). 

The classic initiation mechanism for Ca^2+^ signalling is via IP_3,_ which is produced by the action of a phospholipase C on phosphatidylinositol 4,5-bisphosphate (PIP_2_) [24]. The two classes of “opsonin receptors” both activate PLCγ through tyrosine kinase receptors. These include the Fc receptors, triggered by antibody-opsonised targets and β2 integrin, triggered by complement (C3bi)-opsonised targets [25]. PLC β in neutrophils is also activated by G protein-coupled receptors for soluble agonists, such as formylated peptides. The activation of either PLC generates IP_3_, which diffuses from the plasma membrane away from the initiation site to engage IP_3_ receptors on Ca^2+^ storage sites throughout the cell. This releases Ca^2+^ from the Ca^2+^ stores, which in turn signals plasma-membrane-channel opening through a mechanism known as “store operated Ca^2+^ channel” (or SOCC) opening [26,27]. Unlike Ca^2+^, IP_3_ has a long effective range [19] of 24 μm (having both a higher diffusion constant, 283 μm^2^/s, and having no buffers). Thus, for cells smaller than 24 μm, IP_3_ will fill the cell and act as a global messenger, thus losing any directional information as to the locus of its formation at the phagocytic site (Figure 3).

Route (a) shows the effect of direct coupling of the stimulus to the opening of Ca^2+^ influx channels (here supposed to be the transient receptor potential cation channel, subfamily M, member 2, or TRPM2). Ca^2+^ influx from that site, with limited diffusion distances, would retain information as to its source. In contrast, (b) is the signal for Ca^2+^ via PLC. The generation of IP_3_ with its extensive diffusion would result in the opening of Ca^2+^ channels from remote Ca^2+^ storage sites in the cell, which would open store-operated Ca^2+^ channels (SOCC) in the plasma membrane at all loci and so lose directional information. 

The mechanism by which store-operated Ca^2+^ Channels (SOCC) open when Ca^2+^ is depleted from the endoplasmic reticulum may involve the Orai1 and STIM1 system [28,29,30]. There is considerable evidence that the Orai–STIM system also operates in phagocytes [8,31,32] and is responsible for hotspots of high Ca^2+^ at the formed phagosome–cytosol boundary [33].

A Ca^2+^ influx mechanism which may not involve the depletion of Ca^2+^ stores or IP_3_ generation is the opening of TRPM2 channels (transient receptor potential cation channel, subfamily M, member 2), Ca^2+^-permeable channels which are highly expressed in phagocytic immune cells [34,35,36]. These plasma membrane channels, previously called LTRPC2 or TRPC7, are permeable to both Na^+^ and Ca^2+^ and may be involved in immune-cell behaviour including adhesion and migration [37]. TRPM2 is activated or directly opened by ADP-ribose (ADPR) as well as by oxidants (e.g., H_2_O_2_) [38]. There is evidence that neutrophils (and probably other phagocytes) utilise these Ca^2+^-signalling pathways [39,40,41]. Although the mechanism of activation of the channel or of the generation of ADP-ribose is not fully established, oxidants are known to open the channel directly and also to enhance the opening by ADPR [38]. As oxidants are generated locally within the phagosome, they may play a role in local TRPM2 opening through H_2_O_2_ signalling. The key point, however, is that since TRPM2 activation does not involve IP_3_ generation, there is at least a theoretical possibility that an opsonin-triggered Ca^2+^ signal could remain localised to within a few microns of the phagocytic contact site if TRPM2 channels were involved. 

## 4. Localisation of Ca^2+^ Signalling of Phagocytosis

An early paper [42], using a single-wavelength Ca^2+^ reporter, Quin2, reported that Ca^2+^ signalling was highest in the pseudopodia and periphagosomal cytoplasm. Although fura2 was reported to give a similar result [43], persistent, localised, elevated Ca^2+^ during pseudopodia formation and the onset of phagocytosis cannot generally be confirmed using modern dual-wavelength or confocal imaging of a number of probes. The reported Quin2-elevated fluorescent signals in pseudopodia (i.e., prephagocytic structures) were probably the result of an optical effect whereby fluors are more efficiently excited and emission-detected in the less light-scattering cytoplasm of pseudopodia [44]. When followed in time, the Ca^2+^ signal which accompanies phagocytosis, although presumably triggered locally, causes a global elevation in cytosolic Ca^2+^ (14,16, 45). In one study [45], the question was addressed of whether the presence of the Ca^2+^ indicator itself affected the Ca^2+^ distribution by being an additional Ca^2+^ buffer with high mobility. The Ca^2+^ indicator (fluo4) was, therefore, chemically coupled to a large protein with low mobility and microinjected into neutrophils before phagocytosis. This indicator also failed to detect localised Ca^2+^ signalling. How, then, can Ca^2+^ signal the localised events that precede and trigger phagocytosis? The most likely explanation is that the protein on which Ca^2+^ acts has a low affinity for Ca^2+^, so that only protein close to the source of Ca^2+^ signalling will be activated, as illustrated in Figure 1.

With this in mind, it is relevant that one of the many cytosolic targets of an elevated Ca^2+^ level has a very low affinity for Ca^2+^, the Ca^2+^-activated protein calpain. There is increasing evidence that calpain activity is involved in the formation of phagocytic pseudopodia [46]. For example, pharmacological inhibition of calpain slows phagocytosis by arresting the rapid phase of internalisation [13,15]. The outcome of calpain activation is the cleavage of ezrin, a protein which “staples” the cortical actin network to the plasma membrane [47], maintaining cell-surface wrinkles, which are a reservoir of extra membrane required for phagosome formation [47,48,49], and also regulates the cell-surface membrane tension [48,49,50]. After ezrin cleavage, there are fewer cell surface wrinkles as measured by SEM [51] or sub-domain FRAP (fluorescence recovery after photobleaching) [52] and, consequently, more membrane available for the formation of the phagosome [45]. Using constructs of ezrin with fluors attached to either side of the cleavage site (each with different excitation and emission properties), the cleavage of ezrin could be monitored by separation of the two fluors [53]. This method demonstrated that ezrin cleavage occurred in the phagocytic pseudopodia as it encroached around the target for internalisation [53]. The release of the plasma membrane from the actin cytoskeleton has also been observed in other cell types forming “cell protrusions”. Membrane-proximal F-actin, i.e., actin, held close to the membrane by crosslinking proteins was reduced at the loci of these cell protrusions in RPE-1 and HUVEC cells [54] and ezrin was decreased at similar protrusion sites in osteosarcoma cells [55]. Furthermore, in response to a photoactivatable ezrin phosphatase used experimentally to reduce the effectiveness of ezrin linking, the cell protrusion rate was increased, suggesting the release of a brake of the protrusion system [55].

While most Ca^2+^-activated cytosolic signalling proteins have kds between 0.1 μM and 1 μM, calpain-1 requires a Ca^2+^ level of 50–100 μM for activation [56]. For example, calmodulin, the ubiquitous Ca^2+^-regulated protein, has a kd for Ca^2+^ of about 0.3 μM, meaning that at the resting cytosolic Ca^2+^ level of 0.1 μM, calmodulin is not active, but when cytosolic Ca^2+^ rises to 1 μM, calmodulin becomes almost fully active. In contrast, calpain-1 has a kd for Ca^2+^ of about 30 μM, so that calpain in the bulk cytoplasm will remain inactive during the Ca^2+^ rise to 1 μM. The consequence of this is that calpain can only be activated very closely to the open Ca^2+^ channel or in a cell location where such a high Ca^2+^ level may be reached. One such location is within wrinkles of the plasma membrane, structures such as microridges. Phagocytes by necessity have such structures as these provide the membrane reservoir needed for the formation of the phagosomes [45,51,52]. It has been estimated that the neutrophils’ surface area has an excess of membrane, two to three times that required simply to enclose the cell volume [45,50]. From the modelling of Ca^2+^ influx into a confined volume of cytosol, it has been estimated that the Ca^2+^ concentration in the cytosol within the wrinkled membrane can reach hundreds of micromolar [57]. This is because open Ca^2+^ channels allow Ca^2+^ to enter the small volume of cytosol within the wrinkle at a faster rate than it can diffuse into the bulk cytosol through the “mouth” of the wrinkle [47,57]. Once the local Ca^2+^ buffers in the wrinkle space are overwhelmed, the free Ca^2+^ concentration can rise to very high levels. The intra-wrinkle Ca^2+^ has been recently measured using a construct, called EPIC3, from a low-affinity genetic Ca^2+^ indictor, CEPIA3 (kd for Ca^2+^ ≈ 11 μM), coupled to ezrin, which locates within the wrinkles and near the plasma membrane [58]. Upon phagocytosis, the intra-wrinkle Ca^2+^ was estimated to rise to 30–80 μM [58], sufficient to activate calpain, release ezrin, and uncouple the plasma membrane from the underlying actin cytoskeletal network [53,58].

## 5. Localisation of Ca^2+^ Signals after Phagocytosis

In contrast to the role played by Ca^2+^ in initiating phagocytosis, after a closed phagosome has formed, elevated periphagosomal Ca^2+^ may occur. It is possible that the phagosome itself provides the localised source of Ca^2+^ for this effect. This periphagosomal Ca^2+^ signal has been reported using the dual-excitation-wavelength probe, fura 2 [59], and more recently using the genetic Ca^2+^ reporter, GCaMP3, where a long-lived halo of elevated Ca^2+^ was detectable lasting over 6 min after phagocytic-cup formation [60]. Reported focal hotspots of Ca^2+^ (Ca^2+^ puffs) near phagosomes have also been visualised [33], which aid lysosome–phagosome fusion and also “boost phagocytosis” by recruiting Ca^2+^ stores near phagosomes [61].

The ion channels through which Ca^2+^ ions leak into the cytosol from the phagosome are probably TRPM2. These have been shown to be on the phagosomal membrane [62] and can be experimentally opened via the intraphagosomal microinjection of ADPR or H_2_O_2_ [62]. Since the intraphagosomal environment is oxidising (as a result of the NADPH oxidase) and contains H_2_O_2_ (the substrate for phagosomal myeloperoxidase), it seems likely that the channels are open and generate the elevated periphagocytic Ca^2+^ and periphagosomal Ca^2+^ hotspots observed. The roles of TRPM2 opening on the plasma membrane and on the phagosome are unclear, but since TRPM2 deletion increases susceptibility to polymicrobial sepsis without reducing phagocytosis itself [63], it would seem more likely that they have a role in postphagocytic events (e.g., lysosome fusion), rather than in signalling phagocytosis. It is obvious, in any event, that since these are postphagocytic events, they cannot be involved in signalling the phagocytosis event. 

## 6. Membrane Lipid Signals

In addition to signals which are generated locally and diffuse into the cytosol, such as Ca^2+^ and IP_3_ as discussed above, there are significant localised changes to the lipid composition of the membrane that form the phagocytic cup and the phagosome. Although phosphatidylserine has been implicated [64], the phosphatidyl inositol products have a clearer role as “membrane organisers” [65,66] and signalling [67,68,69]. Phosphatidylinositol 4,5-bisphosphate (PIP_2_) is also elevated in the phagocytic cup of macrophages and in phagosomes [70,71]. However, in human and murine neutrophils, its concentration decreases rapidly during phagosome formation and is undetectable in the phagosome [72]. In the neutrophil, it seems probable that the decrease in PI(4,5)P_2_ is simply the result of the activity of PI3K and PLCγ, both of which use PI(4,5)P2 as the substrate, the first generating PI(3,4,5)P3 and the second generating IP_3_. It remains uncertain why macrophages and neutrophils differ with respect to PIP2 in the formed phagosome, but the initial rise in PI(4,5)P_2_ may be important in the initiation of the cup-forming stage of phagocytosis, which occurs before opsonin receptors have engaged in sufficient number to trigger a Ca^2+^ signal. An atypical ITAM sequence in moesin has been shown to transduce phagocytic signals without receptor activation [73]. Unopsonised particles which deform the plasma membrane so as to form an artificial phagocytic cup do not induce conventional signalling events, but result in phosphatidylinositol 4,5-bisphosphate (PIP2) accumulation at the contact site. Consequently, the FERM domain of moesin translocates Syk to the membrane via the ITAM motif [70]. Perhaps the PIP_2_ accumulation in the phagocytic cup results from this “primitive” nonreceptor signalling which precedes the triggering of a Ca^2+^ signal for large phagocytic targets. It has also been shown that mice neutrophils which do not express moesin have a defect in adhesion and spreading on to the endothelium (a type of frustrated phagocytosis) in vivo [74], whereas ezrin-deficient cells have no such effect. Although the effects on phagocytosis have yet to be reported, the obligatory nature of moesin would be consistent with its role in phagocytosis as a response element essential for later signalling.

Products of phosphatidylinositol 3-Kinase, i.e., phosphatidylinositol 3,4,5-trisphosphate (PIP_3_), phosphatidylinositol 3,4-bisphosphate (PI(3,4)P_2_) [75,76,77], and phosphatidylinositol 3 phosphate (PI(3)P) [78], are similarly generated and localise to the phagocytic cup or the phagosome. During phagocytosis, the generation of these signals is triggered locally to the contact site between the phagocytic target and receptors on the plasma membrane but are restricted to the membrane of the phagocytic cup and the forming phagosome.

The laws of diffusion outlined in the earlier section also apply to lipids. In other words, free signalling lipids in the membrane tend to diffuse AWAY from the point of its generation at the phagocytic initiation site. The lateral diffusion of lipids in the plasma membrane is slower than ions (such as Ca^2+^) in the cytosol. For example, the fluorescent carbocyanine dyes DiIC_18_(3) or DiIC_16_(3), that insert their two long hydrocarbon chains (of 18 or 16 carbons, respectively) into the lipid bilayer of the plasma membrane, are free to diffuse in pure lipid bilayers, but their diffusion constant is only 9.8 μm^2^/s [79]. In living phagocytic cells (neutrophils), this rate of diffusion is reduced further to only 1.43 μm^2^/s [52]. Presumably, the presence of nonlipid molecules in the plasma membrane (i.e., proteins, receptors, channels, etc.) impede lipid diffusion by forcing the random Brownian motion of individual lipid molecules to follow longer “tortuous paths” [7]. Biological signalling lipids have smaller diffusion constants than artificial fluorescent markers (e.g., DiI) but in the same order of magnitude. For example, the diffusion constant for PIP_2_ in the membrane of living cells is reported to be D = 0.8 μm^2^/s [80] or, by single particle tracking, 0.3 μm^2^/s [81]. PIP_3_ diffuses with a lateral diffusion constant of 0.9 μm^2^/s [82]. Despite small variations in the reported D values, these signalling lipids all have reported diffusion constants in the range 0.1–1 μm^2^/s, and for illustration of the problem, the approximate midpoint D value of 0.5 μm^2^/s will be used. 

Thus, the diffusion length of the signalling lipid along the inner leaflet of the cell membrane from the point of generation at the site of contact with the phagocytic target would be 2 √(Dt) [7] which is about 1.4 μm after 1 s (taking D to be 0.5 μm^2^/s), but during the time required for phagocytosis to complete (50–100 s), the signalling lipids may diffuse up to 14 μm (Figure 4). Through free diffusion, the lipid signal would be significantly larger than the initial contact and great than the membrane area required for internalisation. Even a large experimental target such as zymosan, with an approximately 3 μm diameter, would only have a circumference of only 9.4 μm and the phagocytic cup holding half the particle would have a length of 4.7 μm (see Figure 4). 

The three panels represent the time course of phagocytosis from pre-contact of the particle with the phagocyte to cup formation and nearing phagosome closure. The target is large, having a diameter of 3 μm and circumference of about 9 μm. When the cup has formed, a phosphatidylinositol kinase adds a phosphate to form a signalling lipid such as PI(3,4,5)P3 or its degradation, but signalling, product PI(3,4)P2. The activated kinase at the initial contact point will be less than 5 μm from the edge of the cup, and the distance travelled by the lipids is shown in the last panel. Clearly, the formation of the extending pseudopodia and enclosure of the particle within a closed phagosome must be faster than the diffusion of the lipids from their source; otherwise, the lipids would leak into the surrounding plasma membrane. In the illustration, the activation of the kinase is shown by the asterisk, labelled “PI-kinase”. This is deliberately vague so as not to specify which kinase or which signalling lipid is generated. The product is labelled PIPx, where x is the number of phosphates on the inositol ring and could be one, two, or three.

Thus, it is possible that the lipid-signalling message generated from the centre of the particle would spill over the edges of the cup within 13 s. However, the lipid signals remain spatially restricted to the phagocytic cup and the forming phagosome [70,71,83]. Additionally, unlike free diffusion, the boundary of the signal has a sharp cut off at the edge of the phagocytic cup [75,83]. Following fluorescent PIP3 and PI(3,4)P2-binding proteins shows that from a single point near the phagocytic cup edge, PIP3 is generated and moves around the phagocytic cup at an approximately uniform rate [83], unlike diffusion which slows with distance travelled. Additionally, there appears to be barriers to diffusion from one phagocytic cup to the next when they form around two zymosan particles [83]. Interestingly, when two phagocytic targets are simultaneously available to the cell (either adjacent or at either end of the cell), PIP3 is generated only in one phagocytic cup at a time [83]. Its exact role remains unclear, but the inhibition of Pi3kinase inhibits both phagocytosis and Ca^2+^ signalling. 

From the above, it can be seen that lipid-signalling localisation is achieved by mechanisms other than simple diffusion. Although it is not yet established what the mechanism for the restriction of diffusion might be, several possibilities exist which are discussed below.

## 7. Localisation of Lipid Signalling of Phagocytosis

Perhaps the simplest solution is to consider that the signalling lipids on the inner leaflet of the plasma membrane become bound to immobile elements in the cytosol. The ERM proteins ezrin, radixin, and moesin bind to both PI(4,5)P2 in the membrane and immobile f-actin in the underlying cortical cytoskeleton [84,85]. Thus, PI(4,5)P2 is an anchorage point for ezrin to bind to the immobile polymerised actin, which would consequently hold the lipid locked at points of cytoskeletal crosslinking. Analysis of the diffusion of individual PI(4,5)P2 markers shows that two out of every three PI(4,5)P2 molecules are immobile but that the other third are free to diffuse even where PI(4,5)P_2_-dependent complexes occupy the plasma membrane [82]. A similar explanation may apply to PIP_3_, which also associates with areas of actin polymerisation [86,87]. However, while the immobility of PI(4,5)P2 and PIP3 would explain why these lipid signals become trapped in the actin-rich pseudopodia forming the phagocytic cup, this explanation for the accumulation of PI(3,4)P2 in the phagocytic cup would be a consequence of cup formation rather than the signal for its formation.

A second possibility for the restriction of PIP2, and perhaps other lipid signals, is that diffusion is limited to the phagocytic cup because signalling molecules are prevented from escape. Such an impedance to diffusion could be a “fence” of plasma-membrane-associated proteins at the mouth of the forming phagosome which prevents PIP2 and PIP3 escape [88]. Alternatively, the sharp curvature of the lipid bilayer at the mouth of the forming phagosome [89] may prevent lipid diffusion past that point. It is interesting that when actin polymerisation has been inhibited, PIP3 formation can be triggered by using micromanipulation to push a particle against the cell to form a shallow depression rather than a well-defined phagocytic cup. Under these conditions, PIP3 is not limited to the edges of the deforming particle but leaks into the surrounding plasma membrane [83]. Perhaps the shallow cup lacks either or both the protein “fence” or a sufficiently tight lipid bend to prevent diffusion of the lipid from the phagocytic cup. 

There is also the possibility that diffusion is limited by lipid domains in the plasma membrane itself. Cholesterol-enriched rafts might restrict the free diffusion of PIP2 such that the metabolism of PI(4,5)P_2_ is compartmentalised into plasma-membrane-ordered/raft domains [90,91]. Since PIP2 can reside in such rafts, its concentration may be elevated locally and independently regulated, and may, thus, trap PIP2 within them [90].

The third possibility is that diffusion is restricted by the degradation of the signalling lipid (Figure 5). PI(3,4,5)P3 is produced by the phosphorylation of phosphatidylinositol 4,5-bisphosphate (PIP2) by phosphoinositide-3-kinase (PI3K) and is destroyed by removal of the phosphate by PIP_3_-specific 5-phosphatase enzymes SHIP1 and SHIP2 generating PI(3,4)P_2_ in the plasma membrane or phagosome which is then destroyed by PTEN. The half-life of PIP3 has been estimated as less than 10 s [92]. If the average lifetime of an individual molecule of PI(3,4,5)P3 is, thus, 10 s before it encounters the phosphatase, then the diffusion length (2 √(Dt) [7]) for an individual molecule is expected to be only 4.4 μm. For larger phagocytic targets, this could sufficiently restrict the lipid to provide a truly localised signal. However, for small targets, such as bacteria (e.g., 1 μm in length) this may not be a tight enough signal.

The signalling lipid PIPX is here shown with three phosphates (red spheres), i.e., PIP3. If unimpeded, the signalling lipid would freely diffuse for the diffusion length, d1. However, when it encounters a phosphatase, phosphates are removed (e.g., PI(3,4,5)P3 becomes PI(3,4,)P) and the diffusion length of PIP3 is reduced to d2 by the encounter.

A next possibility is one that is not often discussed, but whereas the smooth membrane surrounding the phagosome provides an ideal diffusion platform, the wrinkled and undulating plasma membrane forces molecules to travel a longer diffusion pathlength (in the z direction) for the same apparent distance from the phagosome (in the xy plane). The effect can be demonstrated using DiI, the membrane fluorescent probe, and has been used to measure the wrinkledness of the membrane during phagocytosis [52]. However, it is unlikely that this effect could account for the observed restricted diffusion of PIP2 and PIP3 near the forming phagosome, as the extending pseudopodia may also be less wrinkled, and the effect is unlikely to be quantitatively large enough.

The final possibility is that either (i) each of the previously discussed possibilities, while not individually able to provide a full explanation, may together cause the effect observed; or (ii) another phenomenon will be discovered which explains the effect.

## 8. Localisation of Lipid Signals after Phagocytosis

Once phagocytosis is complete, the lipids on the phagosomal membrane have an important role in directing subsequent events. This is not the main subject of the review which has focussed on the signalling of phagocytosis. However, PI(3)P accumulates on the completed phagosomal membrane [93] and has an important role in the assembly of the oxidase on the phagosomal membrane, by binding the PX domain of p40*^phox^* with high affinity [93,94,95]. It is, therefore, a key step in the assembly of the NADPH oxidase on the phagosome and for the generation of intraphagosomal reactive oxidants. A PX domain on p47phox also binds to PI(3,4)P2 [96], suggesting that this signalling molecule (a metabolite of PI(3,4,5)P3 which persists on the phagosome) is also important for oxidase assembly. It is obviously important that the active oxidase generates reactive oxygen species only within the phagosome, and not indiscriminately around the cell. Thus, it is crucial that the lipids which promote assembly of the oxidase are the restriction to the phagosome.

## 9. Conclusions

This review has attempted to give insight into an important aspect required for a full understanding of phagocytosis. The biophysics of diffusion is presented as it relates to intracellular signalling, and provides a backdrop for interpreting some classic and more novel studies. As a result, some new ideas are presented alongside some of the well-established and older hypotheses. At the time of writing (the end of 2022), some exciting new techniques are being focussed on phagocytosis which will hopefully yield further novel insights. Phagocytosis is, of course, an ancient cell phenomenon, undertaken by single-celled animals and by single cells in higher animals. While evolution has added layer after layer of complexity, at its heart, phagocytosis must have a simplicity that could be exploited by primitive cells with primitive cell biology. Finding that “heart” of phagocytosis may open a deeper understanding of the phenomenon. It is hoped that this problem (and others) will inspire the next generation of cell biologists to be even more creative and use even more imaginative approaches to solve this, one of life’s many mysteries.

## Figures and Tables

**Figure 1 ijms-24-02825-f001:**
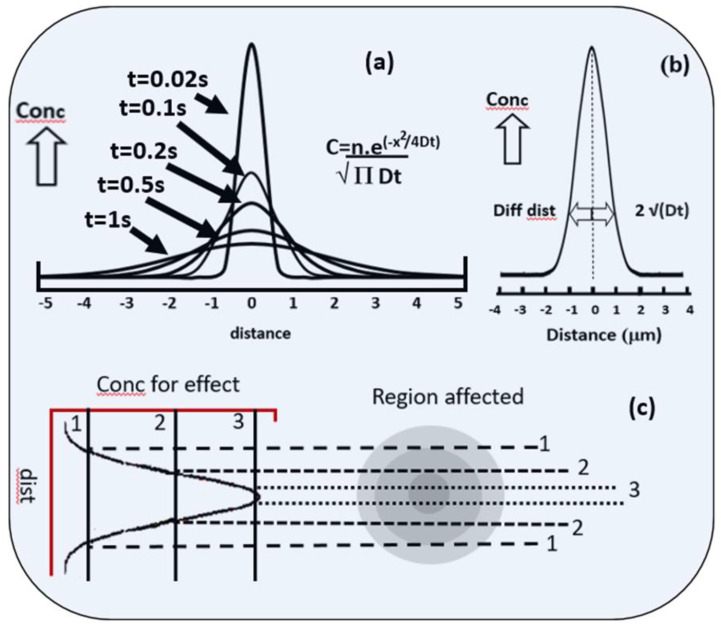
The effect of diffusion of the spread of chemical signals. (**a**) shows the 1-dimenional distribution of the concentration of molecules (or ions such as Ca^2+^) from a point source (at zero distance) with time using the equation shown, where C=1-dimensional concentration (e.g., molecules or ions/μm); n=the number of molecules initially at the source (i.e., at time zero and distance zero): x= distance from the initial source (μm); and D=diffusion constant for the molecule (taken here as 1 μm^2^/s). The concentration of the molecules or ions in the profiles is shown on the vertical axis in arbitrary units. Each curve shows the concentration profile at the time indicated. (**b**) The concentration profile of the signalling molecules at a single time point is shown, with the “diffusion length” (=2 √(Dt) shown for comparison. (**c**) The effectiveness of the signalling-molecule concentration depends on the target requirement, whether it is activated by a low concentration (shown as 1), medium concentration (shown as 2), or high concentration (shown as 3). The “zone of signalling” is reduced by an increased requirement of the target for the signalling molecule as shown by the 1-dimensional spread of activated target molecules (dotted lines) and a 2-dimensional “target” representation, where the darker shading indicates more signalling effectiveness.

**Figure 2 ijms-24-02825-f002:**
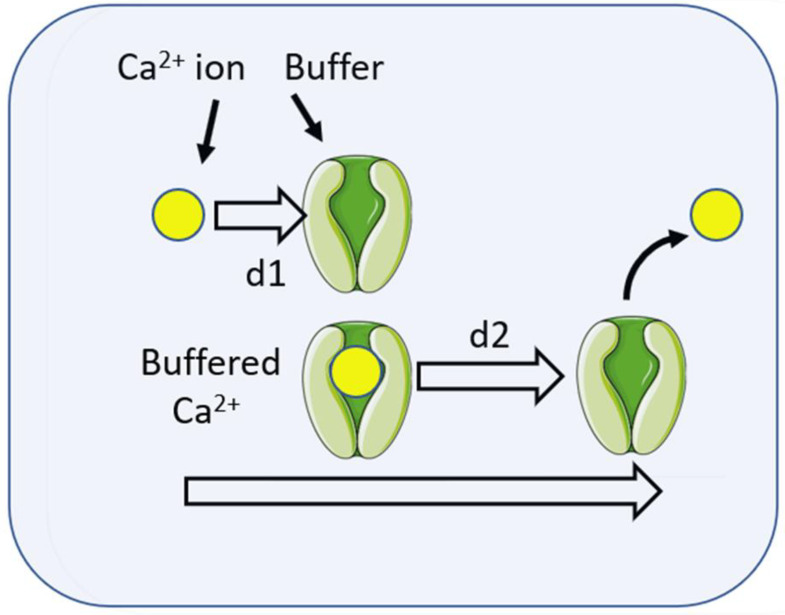
The Effect of Ca^2+^ buffering on diffusion length (signalling reach).

**Figure 3 ijms-24-02825-f003:**
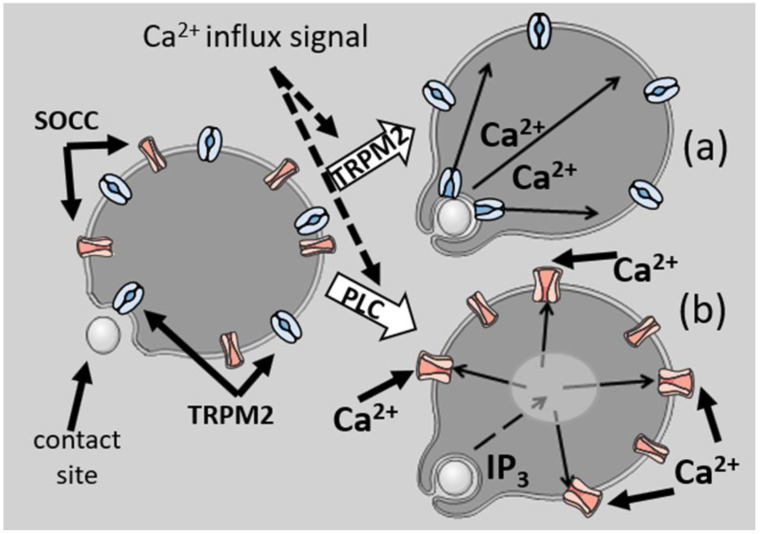
The effect of the two routes of Ca^2+^ signalling on retaining information as to its locus of origin.

**Figure 4 ijms-24-02825-f004:**
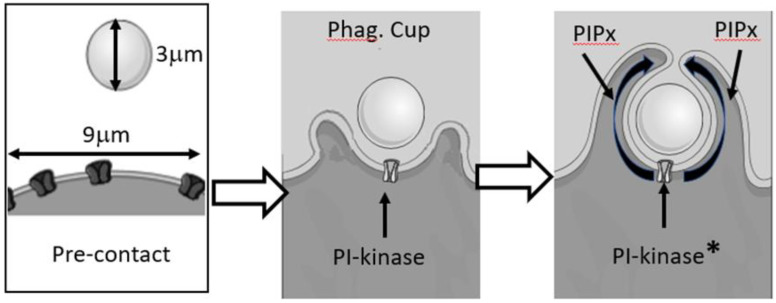
The lateral diffusion of signalling lipids at the site of phagocytosis.

**Figure 5 ijms-24-02825-f005:**
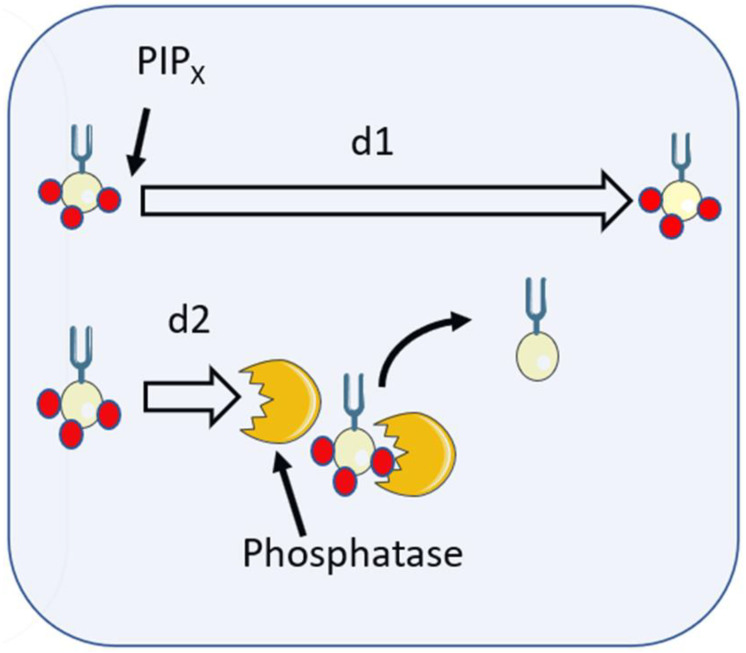
PIPx interactions with inositol phosphatases limit diffusion length.

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
