# Peer review of "Localisation of Intracellular Signals and Responses during Phagocytosis"

_ijms, 2023, doi:10.3390/ijms24032825_

Round 1

Reviewer 1 Report

This article review examines how a common process, as is phagocytosis, is focussed at just one part of the cell were phagosome forms.  To understand the molecular complexity of this process and the mechanism for restriction of signalling the review discusses how diffusion into the cytosol of chemical intracellular signals, in particular Ca2+,  and of signalling lipids along the plasma membrane remain localized. In the first part is discussed if the diffusion of “pre-phagocytic signals, generated by opsonins, can be prevented. In the second part of the review, it is discussed why enzyme-containing granules fuse with phagosomal membrane, where also NADPH oxidase is assembled, but not with the plasma membrane.

Line 48 Instead.

Line 85 Legend to FIG 1 What does “chemical cells” mean?

Line 88 what kind of equation is shown in FIG1? It is not sufficiently explained

Line 91 medium (s) it's actually (2)?

Line 100 useful.

Line 101 These principles apply to... The sentence is not clear

Lines 113-114 The spreading of neutrophils onto surfaces, which is considered to be a “frustrate phagocytosis”, what is meant by “spreading of neutrophils onto surfaces”?

Line 134 2,3 mm is correct?

Lines 135-136 and “effective range” of 135 buffered Ca2+ is about 5mm?

Line 137 and thereafter “Ca2+ buffering content” it would depend on the buffer proteins, give some examples of it.

Lines 139-140 “lysis of lysosomes using GPN, generated transient Ca2+ puffs with diameters of about 3mm” : I can't follow the reasoning well because: the author reports a puff diameter of 3 mm, lysosomes are not the main Ca2+ depot in the cells it must be explained why lysosomes are considered important Ca2+ stores in neutrophils, what does the acronym GPN mean?

Lines 142-143 “However, it depends on the mechanism underlying the Ca2+ signal.” Also this sentence generates a consideration: where does the Ca2+ necessary to activate phagocytosis come from, from intracellular stores or from the outside? The description of Ca2+ response following opsonin signal is discussed later in the text (lines 157-159), it would be better to anticipate the description.

Lines 147-148 in addition to d1 and d2 it is also relevant the dissociation of Ca2+ ions from buffer proteins, given by Kd.

Line 151 “IP3, which produced” and “an phospholipase C” the sentences are not correct

Lines 153155 “b2 integrin, which the signal complement...” the sentence is not clear

Line 193 Legend to FIG3 the acronym (TRPM2) should be explained at the first time.

Lines 193-194 “Ca2+ influx from that site. With limited diffusion distances) would retain information as to its source”. The typing is not correct. In FIG 3 SOC activation is not shown.

Line 211  “may utilisation” is not correct

At line 192 the author speaks about a direct opening of the channel (TRPM2) following the stimulus but further on (209-210) they say that TRPM2 is activated by ADPR, oxidants or arachidonic acid that are generated by precise mechanisms, not only at the activation point.

Lines 218 and 220 report different results for fura2 (or Fura2?)

Lines 230-231 “localised signalling localised Ca2+” it is not a good grammatical construction

Lines 231-232 “trigger phagocytosis.” . is probably ?

Line 295 “opened intra-phagosomal” do you mean “opened by..?”

Line 308 “diffuse into the cytosolic” do you mean “..into the cytosol?”

Lines 317-319 The consideration done for neutrophils is correct so one might add that it would be more appropriate to ask why in macrophages PIP2 remains high in phagosomes

Line 331 “have yet to reported” it should be better ...to be reported

Line 341 “The laws of diffusion outline in the..” it should be better ...outlined

Lines 343-344 “The lateral diffusion in the membrane plasma is slower ions, (such as Ca2+).” The meaning of the sentence is not well understandable.

Line 372 “The three panel” instead of the three panels

Line 376 PI(4,5)P2 is produced by a phosphatidylinositol kinase or is the substrate that generates PI(3,4,5)P3? In addition “PI(4,5)P2 ore PI(3,4,5)P3” correct with or PI(3,4,5)P3

Line 383 “PIPx, where x could be 1,2 or 3.” What 1,2 and 3 indicate?

Line 394 “from one phagocytic to the next” when the author writes phagocytic what does he mean? The phagosome?

Lines 397-398 “PIP3 was generated occurs only in one phagocytic cup at a time” The sentence is not clear

Line 503 “...is of coursed” do the author means ...of course?

From what is written in the introduction it seems that the review is divided into two parts, the second of which treat the localisation  of “post-phagocytic signals”. However the post-phagocytic signal is only marginally discussed in the last paragraph.

In conclusion the review raises a very interesting question regarding the lack of diffusion of the signal starting from the recognition of the opsonins.

The topic is actual and interesting, the topic is treated in an unconventional and attractive way.

However, in some parts of the article, some sentences are unclear, sometimes it would be necessary to specify the statements reported, not obvious to a non-expert reader of the topic. The writing of the work seems hasty and superficial, there are many typing errors

Text writing requires a more linear description of the events and a precise indication of the acronyms, therefore a careful revision.

Author Response

Thank-you for your very careful proof-reading  and for recognising the topic as interesting, and my treatment as 'unconventional and attractive'.

I agree with you that some sentences, as originally written, were unclear, and have now attempted to improve the clarity of the writing. I have corrected the typing errors you point out (together with other that I have found).

As suggested, I have now given a precise definition of each acronym I have used.

Typos and comments:-

Line 48 Instead.  "." has been changed to ","

Line 85 Legend to FIG 1  "chemical cells" has been corrected to "chemical signals". Hopefully this is now self-explanatory.

Line 88 what kind of equation is shown in FIG1? This equation describes 1-dimenional diffusion, where C=1dimensional concentration (eg molecules or ions/mm); n=the number of molecules initially at the source (ie at time zero and distance zero): x= distance from the initial source ; D=diffusion constant for the molecule (taken here as 1micron2/s).  These definitions are now given in the figure legend. It introduces the concept of diffusion length, which has now been added to part (b) of the figure. 

Line 91  (s) has been changed to (2)

Line 100 "useful." has been changed to "useful, "

Line 101 'These principles' etc is not clear. The sentence has now been adjusted to avoid the word  "principles". A  clearer statement has now been made that  the  "concept  of diffusion length " applies to both lipids and ions.

Lines 113-114  what is meant by “spreading of neutrophils onto surfaces”? When neutrophil sediment onto an experimental surface (or unto the vascular endothelium), they change from a spheroid cells (in suspension) to a flattened form (on the surface). The process is usually called "spreading"  because the cell 'spreads out' to occupy a larger area of the contact surface.

Line 134 2,3 "mm" has been corrected to "μm"

Lines 135-136 The “effective range” has been corrected to  5μm

Line 137 The  term “Ca2+ buffering content” is used as a measure of the bound or buffered Ca2+ content in the cytosol, and is the result of binding to all available appropriate molecules in the cytosol. So (for example) in the resting cell with a Ca2+ buffering content of 100μM, a free Ca2+  concentration of  0.1μM , will result  when for  every free Ca2+ ion there 1000 bound/buffered Ca2+ions. The buffering ratio is thsu 1000:1. For greaterclarity, I have now used only this "buffering ratio" explanation. 

Lines 139-140  The  acronym GPN is now explained. It is a dipeptide glycyl-l-phenylalanine 2-naphthylamide which is cleaved within cathepsin C -containing lysosomes  and is widely used to rupture lysosomal membranes and release its  Ca2+ content. The point of the remark is that the lysosome, being a small localised  source of Ca2+,  when released into the cytosol generates a Ca2+ puff with diameters of about that expected from the diffusion length calculation for buffered Ca2+.   I have now included this explanation in the text.

Lines 142-143 where does the Ca2+ necessary to activate phagocytosis come from... it would be better to anticipate the description." Thank-you. I agree that for those new to this area, a simple statement may help them at this point. I have now included this here.

 Lines 147-148 In addition to d1 and d2 it is also relevant the dissociation of Ca2+ ions from buffer proteins, given by Kd.  The kd (dissociation constant) is the  concentration of a molecule which  half-saturates it binding ligand. This is when the rates of association (kon ) and dissociation (koff) are in balance. The time between the Ca2+ buffer binding a Ca2+ ion and releasing it is governed by its "off rate" only. This has now been included 

Line 151 “IP3, which produced” and “an phospholipase C”  has been corrected to "IP­3, which is produced by the action of phospholipase C "

Lines 153155 “b2 integrin, which the signal complement...” has been corrected to "beta2 integrin, which  signals complement .."

Line 193 Legend to FIG3 the acronym (TRPM2) should be explained at the first time.  Its full name is "transient receptor potential cation channel, subfamily M, member 2" has now been included.

Lines 193-194 “Ca2+ influx from that site. With limited diffusion distances) would retain information as to its source”. This has now been corrected to:"Ca2+ influx from that site, with limited diffusion distances,would retain information as to its source."

In FIG 3 SOC activation is not shown. The figure has been redrawn to show SOC channels.

Line 211  “may utilisation” has been corrected to "may utilse"

At line 192 the author speaks about a direct opening of the channel (TRPM2) but TRPM2 is activated by ADPR, etc . This is a good point, which I failed to address properly. The experiments with isolated phagosomes show that microinjection of H2O2 into the phagosome causes opening of TRPM2 channels. Since the NADPH oxidase (which physiologically generates H2O2)  is localised to the phagosome region, I have speculated that TRPM2 opening may also be localised. As it is only speculation, removed it from the original submission. However, in order to answer your point I have rewritten this to suggest that oxidase signally may be important. Thank-you for bringing this to my attention  

Lines 218 and 220  The consensus is that fura2 does not detect localised Ca2+ signals in neutrophils, but I felt it proper to include this a paper,which reported localised signalling using fura2 . I have corrected "Fura" to "fura".

Lines 230-231 “localised signalling localised Ca2+” has been corrected to "to detect localised Ca2+signalling "

Lines 231-232 “trigger phagocytosis.” . is probably ? corrected

Line 295 “opened intra-phagosomal” has been corrected to  “opened by..”

Line 308 “diffuse into the cytosolic” has been corrected to  “..into the cytosol”

Lines 317-319 The consideration done for neutrophils is correct so one might add that it would be more appropriate to ask why in macrophages PIP2 remains high in phagosomes. This is another very good point. I suggested PIP2 accumulation may be part of the "primitive" mechanism. I  have now added a bit more on this.

Line 331 “have yet to reported” has been corrected to "yet to be reported

Line 341 “The laws of diffusion outline in the..” has been corrected to "outlined"

Lines 343-344 “The lateral diffusion in the membrane plasma is slower ions, (such as Ca2+).” has been corrected to "The lateral diffusion of lipids in the membrane plasma is slower than ions, (such as Ca2+) in the cytosol.”

Line 372 The "s" has been added to "panel".

Line 376 PI(4,5)P2 is produced by a phosphatidylinositol kinase or is the substrate that generates PI(3,4,5)P3? In addition “PI(4,5)P2 ore PI(3,4,5)P3” correct with or PI(3,4,5)P3. Phosphatidylinositol 3 kinase adds a phosphate to the 3 position to PI(4,5)P2 to form PI(3,4,5)P3.  The 5-phosphatase removes the phosphate in the 5 postion to leave a different PIP2, namely  PI(3,4)P3. I have now clarified this in the figure legend.

Line 383 “PIPx, where x could be 1,2 or 3.” What 1,2 and 3 indicate? x indicates the number of phosphates ie PIP3,PIP2 or PIP (all with a phosphate on the 2-position.

Line 394 “from one phagocytic to the next” has been corrected to "“from one phagocytic cup to the next"

Lines 397-398 “PIP3 was generated occurs only in one phagocytic cup at a time” has been corrected to "PIP3 was generated only in one phagocytic cup at a time

Line 503 “...is of coursed” has been corrected to "of course"

Reviewer 2 Report

This is an excellent topic discussed in a comprehensive manner. This topic is of great importance to the overall immunology community.

While the structure of the manuscript is well organized, there is a need for careful grammar and punctuation editing throughout the manuscript. While all these are minor points, they detract from the paper's main objective and are very easy to fix. Some of the examples are given below: 

Page 2, line 48: there is a period after the word "Instead".

Page 2, line 57: needs a period instead of comma.

Page 3, line 82: "and" instead of "ad".

Page 4, line 91: should this be "2" instead of "s"?

Page 7, line 211: "utilise" instead of "utilisation"?

Page 10, line 324: "site" instead of "sit"?

Page 10, line 341: "outlined" instead of "outline"?

Author Response

Thank-you for  your careful reading of this paper and for your kind general comments. I agree that there was a need for "careful grammar and punctuation editing" of the originally submitted paper. I am grateful for your proof-reading and have  amended the paper as you have suggested.

Page 2, line 48: there is now a period after the word "Instead".

Page 2, line 57: has period instead of comma.

Page 3, line 82:  "ad" has been changed to "and".

Page 4, line 91: "s" has been changed to "2". 

Page 7, line 211:  "utilisation" has been changed to "utilise". 

Page 10, line 324: The "e" has been added to "sit" to give "site".

Page 10, line 341: "outline" has been changed to "outlined".

 Other problems have also been amended, where they have been spotted by me or pointed out by the second reviewer.

Round 2

Reviewer 1 Report

In the corrected version there are no line numbers so it is difficult to indicate the corrections, anyway

Introduction:

complement molecules (i3Cb) which engage iCb/b2 integrin/complement receptor

pag 7, line 6 “extended from d1 to d1.” The sentence is not clear

pag 7, line 10 “length only about 2.3 μm [19]” it should be better “length is only about 2.3 μm [19

pag 19, legend to fig 4 “PI(3,4,5,5)P32 ore it degradation, but signalling, product PI(3,4,5)P33.” The descripton is still unclear

pag 24 4th line from the bottom “persists on the phagosome ) also it important for oxidase assembly” do the author means is important?

1sth paragraph at page 15 describes localization of Ca2+ signal near phagosome, this is inconsistent with the previous paragraph where “However, localised Ca2+ signalling of the during phagocytosis cannot generally be confirmed using modern dual wavelength (eg Fura2) or confocal imaging of a number of probes.” this contradiction needs to be clarified.

The review has been significantly improved, although some points still need to be corrected.

Author Response

1.  "complement molecules (i3Cb) which engage iCb/b2 integrin/complement receptor" has been amended to "complement molecules (i3Cb) which engage iCb receptors or b2 integrin". I hope this is now clearer. Note that with correct font "b" is "Greek beta". 

pag 7, line 6 “extended from d1 to d1.” has been corrected to “extended from d1 to (d1 +d2).”

pag 7, line 10 “length only about 2.3 μm [19]” has been corrected to “length is only about 2.3 μm"

pag 19, legend to fig 4 “PI(3,4,5,5)P3 or it degradation, but signalling, product PI(3,4,5)P2.” has been corrected to “ PI(3,4,5)P3 or its degradation, but signalling, product  PI(3,4)P2"

pag 24 4th line from the bottom “persists on the phagosome ) also it important for oxidase assembly” do the author means is important? Yes. This has been corrected to  "also is important for oxidase assembly".

1sth paragraph at page 15 describes localization of Ca2+ signal near phagosome, this is inconsistent with the previous paragraph where “However, localised Ca2+ signalling of the during phagocytosis cannot generally be confirmed using modern dual wavelength (eg Fura2) or confocal imaging of a number of probes.” this contradiction needs to be clarified.

Thank-you for pointing this out. This has been amended to "Although  fura2, was reported to give a similar result [43],  persistent localised elevated Ca2+ during pseudopodia formation and the onset of phagocytosis cannot generally be confirmed  using modern dual wavelength or confocal imaging of a number of probes."

In the section on Ca2+ near the phagosome, the difference  between the initiation and completion of phagocytosis  is now highlighted by the addition of the following " In contrast to the role played by Ca2+ in initiating phagocytosis, after a closed  phagosome has formed, elevated peri-phagosomal Ca2+  may occur. It is possible that the phagosome itself provides the localised source of Ca2+ for this effect. "